# Advances in the Application of Protein Language Modeling for Nucleic Acid Protein Binding Site Prediction

**DOI:** 10.3390/genes15081090

**Published:** 2024-08-18

**Authors:** Bo Wang, Wenjin Li

**Affiliations:** Institute for Advanced Study, Shenzhen University, Shenzhen 518061, China; 2300393032@email.szu.edu.cn

**Keywords:** protein language model, nucleic acid binding site prediction, feature extraction

## Abstract

Protein and nucleic acid binding site prediction is a critical computational task that benefits a wide range of biological processes. Previous studies have shown that feature selection holds particular significance for this prediction task, making the generation of more discriminative features a key area of interest for many researchers. Recent progress has shown the power of protein language models in handling protein sequences, in leveraging the strengths of attention networks, and in successful applications to tasks such as protein structure prediction. This naturally raises the question of the applicability of protein language models in predicting protein and nucleic acid binding sites. Various approaches have explored this potential. This paper first describes the development of protein language models. Then, a systematic review of the latest methods for predicting protein and nucleic acid binding sites is conducted by covering benchmark sets, feature generation methods, performance comparisons, and feature ablation studies. These comparisons demonstrate the importance of protein language models for the prediction task. Finally, the paper discusses the challenges of protein and nucleic acid binding site prediction and proposes possible research directions and future trends. The purpose of this survey is to furnish researchers with actionable suggestions for comprehending the methodologies used in predicting protein–nucleic acid binding sites, fostering the creation of protein-centric language models, and tackling real-world obstacles encountered in this field.

## 1. Introduction

The interactions between proteins and nucleic acids form the cornerstone of the functionality of numerous proteins across diverse biological activities and processes, including gene expression, DNA replication, signal transduction, chromatin remodeling, DNA repair, and cellular metabolism, all of which are essential for living organisms [1,2,3,4]. Identifying the nucleic acid binding sites of proteins is vital for comprehending biomolecular mechanisms, elucidating protein functionalities, and facilitating the research into and design of innovative drugs. This effort supports advancements in understanding cellular processes and developing targeted therapies [5]. Contemporary experimental methodologies like X-ray crystallography, nuclear magnetic resonance (NMR) spectroscopy, Cryo-EM [6], and laser Raman spectroscopy have been adapted to decode the complex structures of biomolecular assemblies. These methods excel at resolving intricate molecular structures, each offering unique advantages in understanding complex assemblies. However, experimentally identifying nucleic acid–protein binding sites is labor-intensive and time-consuming. In addition, studies in recent years have emphasized the key role of intrinsically disordered protein (IDP) or region (IDR) in protein–nucleic acid interactions. This includes RNA maturation, ribosome assembly, etc. [7,8]. Unlike structural proteins, IDPs and IDRs lack a fixed three-dimensional structure under physiological conditions. This is also difficult to study by means of experimental assays. Despite the generation of extensive protein data through next-generation sequencing, many proteins still lack nucleic acid binding site annotations. Thus, developing novel, fast, and accurate computational methods for the large-scale identification of nucleic acid-binding residues in proteins is highly desirable [9].

Computational methods for protein–nucleic acid binding site prediction (PNBP) can generally be categorized into two major types: sequence-based and structure-based approaches. In the early stage, sequence-based methods encompass a variety of tools, including NCBRPred [10], DNAPred [11], DNAgenie [12], RNABindRPlus [13], ProNA2020 [14], ConSurf [15], TargetDNA [16], SCRIBER [17], and TargetS [18], which predict residues at nucleic acid binding sites using information solely from protein sequences. While sequence data are abundant, binding sites, despite some spatial configuration conservation, are not always easily identifiable at the sequence level, limiting prediction accuracy [19]. Conversely, structure-based methods, including COACH-D [20], NucBind [21], DNABind [22], DeepSite [23], aaRNA [24], NucleicNet [25], GraphBind [9], and GraphSite [26], tend to achieve higher prediction accuracy by incorporating structural information [27].

Despite many attempts and some progress in computational methods, there are still constraints in the utilization of protein information. First, structure-based methods rely on the Protein Data Bank (PDB) [28], which contains the crystal structures of target proteins. However, many protein structures remain unknown, making structural data much scarcer than sequence data. Moreover, the process of experimentally determining protein structures is both time-intensive and labor-intensive. Sequence-based methods heavily rely on evolutionary insights, requiring extensive comparison and alignment with large protein databases. However, these methods perform poorly when predicting orphan proteins that lack similar entries in the database. The extraction of evolutionary features from proteins necessitates a substantial investment of time. IDP and IDR bring unique challenges for PNBP due to their lack of stable structure and high sequence variability [7]. Lastly, it is crucial to note that current methodologies heavily rely on manually curated features to encapsulate structural information and construct predictive models. This approach requires extensive domain knowledge and may fail to capture essential biological features for specific tasks [24]. 

Notably, the realm of protein structure prediction has undergone significant advancements, largely fueled by the groundbreaking application of deep learning techniques. For example, in the structure prediction competition CASP 14 [29], AlphaFold2 [30], and RoseTTAFold [31] made a major breakthrough in protein tertiary structure prediction, providing raw structural data for large-scale PNBP as a reliable alternative to experimental methods. In addition to understanding protein structure, nucleic acid structure is equally critical for elucidating the mechanisms of protein–nucleic acid interactions. Accurate nucleic acid structures can reveal important binding sites and conformational changes that occur upon binding. Significant progress has been made in structure prediction by deploying large-scale pre-trained biological language models through the attention-based Transformer network. Traditional computational methods for nucleic acid structure prediction play a crucial role in this field. Thermodynamic models predict the secondary structure of nucleic acids based on sequence information, calculate the minimum free energy of possible structures, and determine the most stable conformation. Physics-based modeling methods, on the other hand, use fragment assembly and energy minimization to predict nucleic acid structures with high accuracy. The conformational space of nucleic acid molecules can be explored through Monte Carlo simulations, thus enabling the modeling of large and complex nucleic acid structures [32,33,34,35]. In CASP 15 [36], which focuses more on protein complex and RNA structure prediction, Alchemy RNA learns richer sequence information through pre-trained RNA language models (RNA-FM [37]), ranking first among all AI methods. In addition, protein language modeling (pLM) has also achieved great results. ESMFold [38], for example, differs from previous methods by generating position-specific scoring matrices (PSSMs) from multiple sequence alignments (MSAs) using only protein sequences as inputs. This improves the speed of prediction while maintaining high accuracy at the atomic level. In summary, ESMFold [38] surpasses other methods in handling proteins with limited homologous sequences. Besides protein structure prediction, there is evidence that pLMs also perform well in various other predictive modeling tasks, including protein function annotation [39,40], protein design [41,42], and ligand binding prediction [43,44]. This undoubtedly indicates that pLMs have significant potential in the downstream study of protein function and structure. Consequently, numerous researchers are now devoting their efforts to leveraging the capabilities of pLMs for the large-scale and accurate prediction of protein–nucleic acid binding sites.

PLM has demonstrated numerous advantages over traditional methodologies in PNBP. In pLM, self-supervised learning is utilized to obtain protein representations capable of resolving long-range sequence dependencies and better capturing protein structural information [45]. Moreover, these models can learn rich feature representations directly from large-scale protein sequences, eliminating the need for manual feature extraction. Through sequence-level pre-training, pLMs can capture correlations and binding patterns among nucleic acid binding residues in proteins, encoding them as distinctive feature embeddings. To date, several pre-trained pLMs have been proposed, including ESM [38], TAPE [46], ProtTrans [47], SeqVec [48], and ESM-MSA [49]. These models construct a generalizable architecture through large-scale pre-training on protein sequences and extract diverse and complementary features as embeddings. Numerous prediction models utilizing pLM embeddings have been reported in current literature, such as bindEmbed21DL [39], DeepProSite [50], EquiPNAS [27], ULDNA [51], ESM-NBR [52], and CLAPE [53]. These models consistently outperform those without language model embeddings. Furthermore, extensive studies have demonstrated the robustness of pLM embeddings, their ability to reduce dependence on evolutionary information, and their significant improvement in PNBP accuracy. The inclusion of pLM embeddings in feature combinations significantly enhances overall model performance.

## 2. Biological Language Model

Language models (LMs) excel in content-aware data representation, from sequential databases, making them widely utilized in machine translation, question-answering systems, and even extended to applications in computer vision [54]. Encoder architectures for LMs are typically categorized into two main types: examples of recurrent neural networks (RNNs) encompass the long short-term memory (LSTM) architecture [55], while attention-based mechanisms, exemplified by Transformers [56], offer an alternative approach, both renowned for their powerful capabilities. The Hidden Markov Model (HMM), a cornerstone linguistic framework, is widely utilized in the realms of protein homology modeling and searching. Given the parallels between human language and biological languages, LMs have evolved into biological language models. Through transfer learning, biological language models are effectively applied to characterize the downstream structure and function of biological substances [57].

### 2.1. RNNs and LSTM 

RNNs use a cyclic structure as compared to the traditional model where the nodes are not connected within the network layer and hence can handle temporal data. This was first applied in natural language modeling to capture language context and dependencies. RNNs allow the previously hidden layer to be used as an input and will share the parameters of each step and hence can be used to process variable-length sequences of inputs [58]. The fundamental architecture of a neural network comprises distinct layers: input, hidden, and output, as depicted in Figure 1a, offering a structured approach to data processing. A_t_ each timestep t, the input x_t_ ∈ R^l^, the hidden state h_t_ ∈ R^d^, and the output state vector o_t_ ∈ R^d^ are formulated as follows, with superscripts l and d representing the dimensions of input features and hidden units, respectively, as outlined in [57]:(1)$$h_{t}=f(Ux_{t}+Wh_{t-1})$$
(2)$$o_{t}=g(Vh_{t})$$

Equation (1) represents the formula for the recurrent hidden layer, where U denotes the input weight matrix, W represents the weight matrix for the feedback connection from the previous hidden state, and f serves as the activation function that introduces nonlinearity into the model. Equation (2) represents the formula for the output layer, where V represents the output weight matrix and g is the activation function applied to the layer. The hidden layer has two inputs, the first is the product of U and the x_t_ vector, and the second is the product of the states h_t−1_ and W output by the previous hidden layer, and finally together they output the final o_t_. By iteratively substituting Equation (1) into Equation (2), we derive an expression:(3)$$o_{t}=g(Vh_{t})$$
(4)$$=Vf(Ux_{t}+Wh_{t-1})$$
(5)$$=Vf(Ux_{t}+Wf(Ux_{t-1}+Wh_{t-2}))$$
(6)$$=Vf(Ux_{t}+Wf(Ux_{t-1}+W(f(Ux_{t-2}+Wh_{t-3}))))$$
(7)$$=Vf(Ux_{t}+Wf(Ux_{t-1}+W(f(Ux_{t-2}+W(f(Ux_{t-3}+\ldots )))))$$

From the aforementioned, it becomes evident that the output value o_t_ of the recurrent neural network is intricately influenced by the sequence of preceding input values x_t−1_, x_t−2_, x_t−3_, x_t−4_, and so on, for successive input values. This is the reason why the RNN can consider any number of input values in its computations.

For many language models, bi-directional sequence information is required, and a bi-directional recurrent neural network is needed. As shown in Figure 1c, the hidden layer of a bidirectional neural network maintains two distinct values: A, which participates in the forward computation, and A′, which contributes to the reverse computation. The ultimate output value o_t_ is a synthesis of both A_t_ and A′_t_. Its calculation is:(8)$$o_{t}=g(Vh_{t}+V’{h’}_{t})$$
(9)$$h=f(Ux_{t}+Wh_{t-1})$$
(10)$$h’=f(U’x_{t}+W’{h’}_{t+1})$$

From the three formulas mentioned above, it is evident that the forward and reverse calculations do not share weights, indicating that U and U′, V and V′, as well as V and V′ are all distinct weight matrices.

However, in practice, RNNs do not handle longer sequences well, and the training process is susceptible to issues such as gradient explosion and gradient vanishing. These problems can prevent the gradient from being successfully propagated through longer sequences, ultimately hindering the RNNs’ ability to capture information over long distances. Therefore, LSTM networks was developed to solve this problem [55]. 

The LSTM network has three primary inputs: the current input value x_t_, the output value h_t-1_ from the previous timestep, and the vector of memory cells c_t−1_ from the preceding moment. The LSTM has two outputs: the vector of hidden states at the current moment, h_t_, and the vector of states of the memory cells at the current moment, c_t_. Furthermore, Figure 1b illustrates the utilization of σ and tanh, which signify the sigmoid and hyperbolic tanh layers, respectively, within the neural network architecture. The forget gate layer plays a pivotal role in determining which information from the cell state at timestep t should be discarded, and the remaining features are used to calculate c_t_. x_t_ ∈ R^l^, h_t_ ∈ (−1,1)^d^, and f_t_ ∈ (0,1)^d^, and sigmoid is usually used as the activation function.
(11)$$f_{t}=\sigma ({U_{x}}^{\left(f\right)}x_{t}+{W_{h-1}}^{\left(f\right)}h_{\left(t-1\right)}+b^{\left(f\right)})$$

The input gate decides which values to update. This decision is made based on the input data x_t_ and the hidden state h_t−1_, which are processed through a neural network layer. $\tilde{C_{t}}$ ∈ (−1,1)^d^ represents the candidate cell state update, and i_t_ ∈ (0,1)^d^ is the input gate’s activation vector
(12)$$i_{t}=\sigma ({U_{x}}^{\left(i\right)}x_{t}+W_{h-1}{h_{t-1}}^{\left(i\right)}+b^{\left(i\right)})$$
(13)$${\tilde{C}}_{t}=tanh({U_{x}}^{{(\tilde{C}}_{t})}x_{t}+{W_{h-1}}^{\left({\tilde{C}}_{t}\right)}h_{t-1}+b^{\left({\tilde{C}}_{t}\right)})$$

Compared to traditional RNNs, LSTM networks incorporate a unique gate mechanism, known as the cell state, which enables precise control over the flow and retention of features. This cell state is represented by the horizontal line at the top of Figure 1b, which runs through the entire chain-like system. To update the cell state C_t_ to its new value, we combine the old cell state C_t-1_ with the new candidate state.
(14)$$C_{t}=f_{t}⊙C_{t-1}+i_{t}⊙{\tilde{C}}_{t}$$

To obtain the prediction value and prepare the input for the subsequent time step, the hidden state’s output h_t_ is computed by passing it through the output gate, where o_t_ ∈ (0,1)^d^.
(15)$$o_{t}=\sigma ({U_{x}}^{\left(o\right)}x_{t}+{W_{h-1}}^{\left(o\right)}h_{t-1}+b^{\left(o\right)})$$
(16)$$h_{t}=ot⊙tanh(C_{t})$$

### 2.2. Attention Mechanism and Transformer

The traditional Sequence-to-Sequence (Seq2Seq) model uses RNN or LSTM as an encoder or decoder to process the sequence and extract features [59]. However, it is difficult for RNN or LSTM as an encoder to fully retain input sequence information in their final state. In addition, the computation of RNN and LSTM is time-dependent and therefore difficult to compute in parallel, which leads to very slow computation. Vaswani et al. [56] proposed the Transformer model to compensate for these shortcomings. The Transformer model utilizes self-attention, enabling it to individually access and weight all prior states. Furthermore, the Transformer abandons the traditional horizontal RNN transmission and only transmits vertically, requiring only the stacking of self-attention layers. This approach enables parallel computation within each layer and can be accelerated using a GPU. The proposed BERT [60] and GPT [61,62] models based on this perform exceptionally well in natural language processing tasks. Unlike RNNs, the Transformer simultaneously processes the entire input sequence using stacked self-attentive layers in both its encoder and decoder. Each layer includes a multi-head attention module, allowing a residual connection for preventing network degradation, and layer normalization modules for normalizing the activation values of each layer (Figure 2). Within the Transformer, the fundamental single-head attention mechanism is termed “Scaled Dot-Product Attention”, where the self-attention output is derived through a specific computational process.
(17)$$Attention(Q,K,V)=softmax(\frac{QK^{T}}{\sqrt{d_{K}}})V$$

Self-Attention is a fundamental component of the Transformer, as illustrated in Figure 3a. The inputs Q (queries), K (keys), and V (values) for self-attention are linearly transformed from the input matrix X, which comprises vectors x or the output from the preceding encoder layer. Multiple attention consists of combinations of multiple self-attention and allows parallel attention to information from different subspaces (Figure 3b). By directing X through h distinct self-attention layers, h output matrices Z are generated. These matrices are then concatenated and further processed by a linear layer, yielding the final output matrix Z, which mirrors the input matrix X’s dimensions.
(18)$$MultiHead(Q,K,V)=Concat({head}_{1},\ldots ,{head}_{h})W^{O}$$
(19)$$where\;{head}_{i}=Attention(Q{W_{i}}^{Q}K{W_{i}}^{K}V{W_{i}}^{V})$$

In the context of multi-head attention, projections are executed utilizing specific parameter matrices W_i_^Q^ ∈ $R^{d\times d_{i}}$, W_i_^K^∈ $R^{d\times d_{i}}$, W_i_^V^ ∈$R^{d\times d_{i}}$, and W^O^ ∈$R^{{hd}_{i}\times d}$, d_i_ = d/h. This configuration comprises h parallel attention layers, or ‘heads’, each independently processing the input data.

The formula for the Add and Norm layer is as follows: (20)LayerNorm(X+MultiHeadAttention(X))
(21)LayerNorm(X+FeedForward(X))
where X signifies the input data fed into either the Multi-Head Attention module or the Feed Forward module. The respective outputs of these two components are denoted by MultiHeadAttention(X) and FeedForward(X). The operation add refers to adding the input X to the output of the Multi-Head Attention module, resulting in X + MultiHeadAttention(X). This is a form of residual connection, commonly used to address the challenges of training deep neural networks by allowing the network to focus on learning the differences in the current layer. Layer Normalization is a technique commonly employed in RNN and other neural network architectures. It serves to regulate the input to each layer of neurons by ensuring they share a consistent mean and variance. This standardization process aids in accelerating the training convergence by mitigating the issue of internal covariate shift, where the distribution of layer inputs changes during training.

Furthermore, the feed-forward module comprises a two-layer fully connected network. The first layer employs ReLU as its activation function, while the second layer does not utilize any activation function. Furthermore, in addition to embedding sequence information, the Transformer model requires the embedding of positional information at the start of both the encoder and decoder to represent the position of each element in the sequence.

### 2.3. Protein Language Models 

DeepCNF, proposed by Wang et al. [63], utilizes a deep neural network architecture, incorporating LSTM, to predict protein secondary structure (SS). This approach effectively addresses the challenge of long-range dependencies in sequence processing. This method consistently matches or exceeds state-of-the-art models in SS prediction. SPIDER3-Single [64] innovatively eliminates the dependency on evolutionary information, enabling predictions based solely on individual sequences. Other models, such as OCLSTM [65] and LSTM-BRNN [66], share similar research objectives and architectural frameworks. Additionally, models like GLTM [67] and LSTMCNNsucc [68] utilize pre-trained pLM embeddings to tackle diverse downstream tasks, including protein polymer motif prediction and post-translational modification prediction. These studies collectively demonstrate the efficacy of LSTM within pLMs in capturing the intricate biological features of proteins.

Recent advancements have introduced a range of deep neural language models specifically for protein sequences. Notable examples include ESM [69], TAPE-Transformer [46], ProtTrans [47], UDSMProt [70], UniRep [71], and ESM-MSA [49], each demonstrating remarkable progress in the field. ESM2 [38], a key component of the ESM framework advanced by the DeepMind team (Table 1 details the ESM family of models), builds upon the powerful Transformer architecture. This unsupervised deep attention neural network boasts a multi-layered architecture, with each layer integrating multiple attention heads alongside a feed-forward network (FFN). ESM2, with its 15 billion parameters, is trained on approximately 43 million protein sequences, leveraging mask pre-training to learn sequence–structure–function relationships and apply these learned features to downstream tasks. 

ESM-MSA [49], as depicted in Figure 4, excels in unsupervised learning by encoding input MSA knowledge into feature embedding matrices. Each module integrates row-attention and column-attention layers to capture co-evolutionary relationships among amino acids at the sequence and positional levels. Another notable model, ProtTrans [47] (see Table 2), shares a similar architecture to ESM2. Both models demonstrate the capability of pLMs to extract syntactic content from large-scale protein sequences, even without relying on MSA information alone. Furthermore, integrating multiple sources of information such as structure, function, MSA, and other biological priors enhances protein characterization [72].

ProteinBERT [73], another significant model, incorporates Gene Ontology (GO) annotations during pre-training, enriching protein characterization by combining sequence data with GO annotation information to predict protein functions effectively.

**Table 1 genes-15-01090-t001:** ESM family.

Hyperparameter	ESM-1b	ESM-MSA-1b	ESM-1v	ESM-2
Dataset	UniRef50	UniRef50	MSA	UniRef90
Number of layers	33	12	33	48
Params	650 M	100 M	650 M	15 B
Embedding Dim	1028	768	1028	5120
Input	Single-sequence	MSA	Single-sequence	Single-sequence
Universality	Family-specific	Few-shot	Zero-short	Zero-short
Model	Transformer	Two rows of attention mechanisms have been added	Transformer	Transformer
References	[69]	[49]	[74]	[75]

**Table 2 genes-15-01090-t002:** ProtTrans family.

Hyperparameter	ProtTXL		ProtBert		ProtXLNet	ProtALbert	ProtElectra	ProtT5-XL		ProtT5-XXL	
Dataset	BFD 100	BFD 100	UniRef 100	UniRef 100	UniRef 100	UniRef 100	UniRef 100	UniRef 50	BFD 100	UniRef 50	BFD 100
Number of layers	32	30	30		30	12	30	24		24	
Params	562 M	420 M	409 M		409 M	224 M	420 M	3 B		11 B	
Hidden layers size	1024		1024		1028	1024	1024	1024		1024	

**Figure 4 genes-15-01090-f004:**
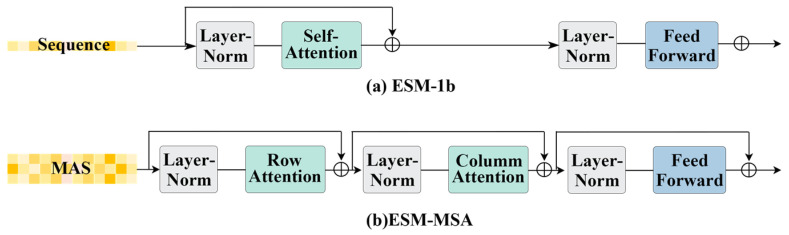
Core modules of ESM-1b and ESM-MSA.

### 2.4. Nucleic Acid Language Models 

In addition to pLM, there have been good advances in language models designed specifically for nucleic acids, including DNABERT [76] and RNA-TorsionBERT [77]. These models are based on the BERT [60] (Bidirectional Encoder Representation from Transformers) architecture and are tailored to capture the unique features and sequence patterns of DNA and RNA.

DNABERT [76] and RNA-TorsionBERT [77] adapt BERT models for DNA and RNA sequences. They are trained on large-scale genomic data to learn the underlying patterns of nucleotide sequences. These models have been successfully applied to a variety of tasks and have helped to deepen the understanding of the PNBP mechanism. It provides a powerful approach to understanding the complex dynamics of protein–nucleic acid interactions. For instance, DNABERT has been used in tasks like identifying transcription factor binding sites and predicting methylation patterns, while RNA-TorsionBERT has been applied to understand RNA conformational dynamics and to predict RNA–protein interactions.

However, in comparison to pLMs, nucleic acid language models have limited training data, which restricts the generalization ability of the models. In addition, the progress of pLMs is due to the rich evolutionary information contained in the protein sequences themselves, whereas nucleic acids themselves may not contain similarly rich information. Especially in non-coding regions and species-specific regulatory elements, it is difficult for nucleic acid language models to obtain better access to evolutionary information [35]. Nucleic acid language modeling is at an early stage of development and needs further validation, but it still has a very promising future.

Looking forward, the future of nucleic acid language models is undoubtedly promising. Continued advancements in genomic sequencing technologies and the accumulation of more comprehensive datasets could potentially address the current limitations. Additionally, integrating nucleic acid models with other types of biological data, such as epigenetic marks, chromatin accessibility, and transcriptional activity, could enhance their ability to make accurate predictions. As these models evolve, they are expected to play an increasingly crucial role in decoding the intricacies of genetic regulation, gene expression, and the broader mechanisms underlying protein–nucleic acid interactions.

## 3. Methods of Nucleic Acid Protein Binding Sites Prediction

This section showcases cutting-edge models designed to predict nucleic acid and protein binding residues, reflecting the latest advancements in current research endeavors. It distinguishes these models and compares them based on their use of pLMs as feature embeddings, highlighting the clear advantages of pLMs in this prediction task. Figure 5 categorizes the features that are currently in common use, and the methods mentioned in the text involve a wide range of features at the sequence and structural level.

### 3.1. Overview of Methods Framework 

A typical generic framework for PNBP consists of three primary modules: the protein feature embedding unit, the backbone network module, and the loss computation module. The embedding unit constructs representations of input proteins leveraging evolutionary data or crafting discriminative embeddings through pLM, while also incorporating structural attributes as salient features. These features are subsequently fed into the backbone network, which then performs PNBP within the input protein. Deep learning has demonstrated significant advantages in predicting nucleic acid–protein binding sites, with many models discussed in this review utilizing various mainstream neural networks such as MLPs, CNNs, RNNs, GNNs, and LSTMs, in addition to traditional machine learning approaches like SVMs.

Finally, the loss calculation module performs backpropagation using diverse loss functions such as binary classification loss, contrastive loss, cross-entropy loss [78], class-balanced focal loss [74,75], and triple center loss (TCL) [79], among others. These functions guide the updating of model parameters based on the calculated loss.

### 3.2. Benchmark Datasets

To ensure fair comparisons between different methods, most models utilize benchmark datasets derived from previous works, such as GraphBind [9], which predominantly builds upon BioLiP [80]. BioLiP is a database of biologically pertinent ligand–protein interactions meticulously curated from the Protein Data Bank (PDB) [28]. It meticulously curates interactions through a blend of computational validation and manual verification, ensuring biological relevance by filtering out non-biologically significant ligands.

Within BioLiP, a residue qualifies as a binding residue if its minimum atomic distance from a nucleic acid molecule falls below a threshold calculated as 0.5 Å added to the combined van der Waals radii of the two closest atoms. This criterion is used to complement experimental data. Each entry in BioLiP is replete with annotations, encompassing details like ligand-binding residues, binding affinities, catalytic sites, enzyme classifications, gene ontology terms, and hyperlinks to related databases, offering a holistic view of the ligand–protein interactions [80]. 

The GraphBind method, for instance, leverages BioLiP’s comprehensive annotation of nucleic acid binding residues. It achieves this by aggregating binding residues across multiple similar or identical protein complexes, where proteins may interact with different DNA or RNA fragments in the collated data. As of 15 June 2024, BioLiP comprised 43,648 DNA–protein complexes and 153,190 RNA–protein complexes.

Xia et al. utilized the BioLiP database for their work, focusing on DNA and RNA binding proteins. They excluded DNA–RNA–protein complexes and divided proteins based on reporting dates, using sequences reported before 6 January 2016, for training. To tackle the imbalance between binding and non-binding residues in the data, they augmented the training set by increasing the number of binding residues using bl2seq [81] and TM-align [82] algorithms for sequence and structure comparisons. They also reduced sequence redundancy using CD-HIT [75] (threshold 30%). Ultimately, their final training set comprised a robust collection of 573 DNA-binding protein chains and 495 RNA-binding protein chains. Similarly, their test set, subjected to comparable processing measures, encompassed 129 DNA-binding proteins and 117 RNA-binding proteins.

Other studies similarly rely on BioLiP-based datasets, employing various data processing techniques such as CD-HIT [83] for sequence redundancy reduction and TM-align [82] for dataset partitioning into training and test sets. Some studies also create validation sets to ensure model robustness and avoid overfitting during hyperparameter tuning.

### 3.3. Feature Extraction

Extracting discriminative features is crucial for accurately predicting nucleic acid–protein binding sites, leading researchers to explore innovative approaches to characterize protein sequences and structures. The widely held conviction underscores the notion that a protein’s sequence serves as the blueprint for its three-dimensional (3D) structure and functionality, thereby fueling the adoption of diverse sequence-level features in relevant analyses. Commonly employed features include amino acid species encoding, residue propensity calculations, and physical properties. Evolutionary information derived from sequence comparisons is also frequently utilized to describe interactions. Moreover, pLMs, highlighted in this paper, effectively extract information from protein sequences as features.

At the structural level, proteins are often characterized by encoding 3-state and 8-state secondary structures (SS), while relative solvent accessibility (RSA) data have proven significant in predicting nucleic acid binding sites [84]. Local geometric features, residue orientation, and other structural attributes further enhance feature extraction in many models. In addition, geometric deep learning is well practiced in protein structure modeling and can be used to extract advanced biophysical–chemical knowledge of the structure.

In summary, a diverse array of features can be harnessed to accurately predict nucleic acid–protein binding sites, combining features from different sources and assigning appropriate weights enriches feature representation, enhancing model performance. By integrating multiple features, both protein structure and sequence information can be fully utilized to achieve more accurate PNBP, thereby improving model robustness and generalization.

#### 3.3.1. Features Based on Amino Acids

Identifying nucleic acid binding residues entails employing various methods that draw upon amino acid composition, residue preferences, and physicochemical characteristics. Amino acid composition analysis determines the relative abundance of each amino acid type around DNA binding sites, providing insights into their prevalence in nucleic acid interactions. Residue propensity calculation assesses the likelihood of specific residues being involved in nucleic acid binding, revealing which amino acids are preferred in these sites. Biochemical composition analysis examines the physicochemical properties of residues—such as polarity, hydrophobicity, and charge—to understand their functional roles in nucleic acid interactions. Patiyal et al. utilized the Pfeature [85] package to compute these features comprehensively [86]. In contrast, GeoBind [87] employs a lightweight neural network that avoids hand-crafted physicochemical descriptors. Instead, it adopts an atomic point cloud approach similar to dMaSIF, where chemical features are succinctly represented as a 1 × 6 vector. This vector employs one-hot encoding to capture the presence of specific atoms, including Carbon (C), Hydrogen (H), Oxygen (O), Nitrogen (N), Sulfur (S), and others, offering a precise and efficient way to encode chemical information. This approach enables GeoBind to effectively analyze protein surfaces by distinguishing atom types, providing crucial information for subsequent prediction tasks.

Pseudo-positions capture the center of mass for each residue, taking into account both main chain and side chain atoms to represent their positions in nucleic acid interactions. This feature type is essential for modeling interactions involving both main and side chain atoms.

Additionally, atomic features describe the physicochemical properties and structural characteristics of residues. Xia et al. [9] extracted seven features for each residue, which encompass atom mass, B-factor, a flag indicating whether it belongs to a side-chain, its electronic charge, the number of bonded hydrogen atoms, ring status, and van der Waals radius. These features were averaged across residues to create an atomic feature matrix (L × 7) for each query protein, where L represents the number of residues.

For simplicity, approaches such as Roche et al. [27] have adopted one-hot encoding as a means to represent the 20 distinct amino acid residue types, facilitating the straightforward integration of amino acid type information in predictive modeling tasks.

#### 3.3.2. Features Based on Evolutionary Information

The PSSMs serve as a valuable tool for representing residue conservation in protein sequences. PSSMs are typically generated using the PSI-BLAST tool [88], which employs heuristic and dynamic programming algorithms to search for homologous sequences in databases like NCBI’s non-redundant (NR) database or Swiss-Prot, with an output size of L × 20.

Research by Xia et al. has demonstrated that different backend algorithms and databases, such as PSI-BLAST and HHblits, yield complementary results [9]. HHblits utilizes a Hidden Markov Model (HMM) to search the uniclust30 database, generating an HMM matrix of size L × 30 [89]. The HMM matrix is structured to encapsulate crucial information pertaining to amino acid sequences. It comprises 20 columns that mirror the observed frequencies of each of the 20 amino acids within homologous sequences. Additionally, seven columns are dedicated to representing transition frequencies and three columns are utilized to encapsulate local diversities.

MSA information plays a crucial role in assessing protein homology. Tools like Clustal Omega [90], MAFFT [91], and MUSCLE [92] offer various algorithms and parameters to enhance accuracy and robustness in comparing protein sequences. MSAs provide insights into structural and functional relationships among proteins, revealing conserved and variable regions and aiding in understanding evolutionary and structural features. Several pipelines facilitate MSA generation, such as the ColabFold [93] pipeline utilized by Roche et al. [25], which employs MMseq2 [94] to generate MSAs from amino acid sequences.

#### 3.3.3. Feature Based on Structure

Protein 3D structures are crucial for revealing nucleic acid–protein binding sites and are frequently employed to predict their characteristics. Solvent accessibility, introduced by Lee and Richards [95], plays a pivotal role in this process because it identifies the protein surface regions likely to interact with other molecules. Solvent accessibility categorizes residues as buried (B), intermediate (I), or exposed (E), and can be accurately predicted using programs like SANN [96]. This method constructs an RSA (Relative Solvent Accessibility) feature for each residue using a sliding window of size 9, resulting in an RSA feature vector of dimension 27. Additionally, RSA can also be computed using the DSSP program [97]. Apart from RSA, calculating the protein’s monothermal secondary structure profile deriving the sine and cosine values of the protein backbone torsion angle ϕ are effective methods for representing the protein’s 3D structure. By leveraging one-hot encoding to represent SS in either a 3-state or 8-state format, each amino acid residue within a protein sequence can be transformed into a multidimensional feature vector, where each dimension corresponds to a specific type of secondary structure. The torsion angles, including the C=O cosine angle between consecutive residues, the torsion angle between consecutive Cα atoms, and the normalized backbone torsion angle, provide critical local geometric information about the protein’s backbone. These angles offer insights into local spatial relationships, torsion, and curvature of the protein backbone, enhancing our understanding of its structural functions.

Predicted protein structures can also serve as features for PNBP, as demonstrated in studies like DNABind [22]. These studies used predicted secondary structures and RSA to characterize residue structures, with secondary structures generated using programs such as SPINE [98] based solely on input protein sequences.

Geometric deep learning plays a crucial and impactful role in extracting features from protein structures. GPSite [99] and GPSFun [100] utilize a geometric featurizer to extract atomic and inter-atomic features, treating residues as nodes and constructing protein radius maps to represent protein structures. Experimental findings have convincingly demonstrated that this approach captures and represents the intricate three-dimensional structural features of proteins.

#### 3.3.4. Feature Representation Extraction from pLMs

While traditional hand-designed features have been effective in predicting nucleic acid–protein binding sites, they are constrained by a priori knowledge and may not capture diverse patterns, limiting their feature extraction capabilities. Moreover, these features often lack generalization ability and struggle to adapt to different prediction tasks. In contrast, pLM embeddings offer a more comprehensive, accurate, and adaptable feature representation. pLM embeddings are learned through large-scale pre-training, making them highly transferable and effective even with limited data samples. When used in analysis, pLM embeddings can be directly integrated into models for end-to-end learning. This simplifies the modeling process, circumvents the complexities of manual feature design, and enhances flexibility and versatility. Furthermore, pLM embeddings can be further harnessed by fine-tuning them for specific tasks or domains, improving their characterization capabilities and overall performance.

ESM2, for example, was trained on the Uniref50 [101] database, leveraging evolutionary insights from millions of sequences. It offers a range of pre-trained models with varying sizes, each providing feature matrices of different dimensions (e.g., 320, 480, 640, 1280, 2560, 5120) for protein sequences of length L. However, due to GPU limitations, processing long sequences (e.g., length > 1000) with even smaller models can still lead to memory issues, necessitating sequence segmentation while retaining sufficient evolutionary information.

Additionally, other methods like ProtT5-XL-U50 have been pre-trained on datasets like BFD [102], yielding 1024-dimensional sequence embeddings normalized to scores between 0 and 1. This normalization ensures consistent scaling across embeddings, facilitating straightforward integration into predictive models using tools like HuggingFace’s Transformers(v4.44.0) [103].

### 3.4. Performance Evaluation

Here we evaluate several methods for predicting nucleic acid binding sites using pLMs as feature embeddings: CLAPE [53], ESM-NBR [52], DeepProSite [50], EquiPNAS [27], ULDNA [51], GPSite [99], and GPSFun [100], which are generally summarized in Table 3. We compare them to cutting-edge methods that rely on manual feature extraction, evaluating their performance side-by-side.

Firstly, compared to sequence-based models, pLM embeddings significantly enhance model performance due to pre-training. Notably, some of these methods outperform most structure-based methods, such as CLAPE [53], even without leveraging any structural information. Meanwhile, the accuracy of protein structures is crucial for prediction tasks, as predicted structures are often less useful for PNBP. When using experimental protein structures, CLAPE’s performance is slightly lower than GraphBind [9]. However, when replacing with predicted protein structures, CLAPE [53] outperforms GraphBind [9], highlighting the superiority of pLMs in the absence of precise structural data. In experiments with the EquiPNAS [27] method using structural features and pLM embeddings, EquiPNAS [27] outperforms other methods even when using AlphaFold2-predicted structures as inputs, and the performance degradation is small compared to using experimental structures. The embeddings from pLMs undoubtedly contributed significantly to this performance improvement. GPSite [99] and GPSFun [100] extract features from structures predicted by ESMFold using geometric characterizers and incorporate pLM as embeddings. This approach leverages the accuracy and computational efficiency of language models, resulting in prediction accuracies that surpass most experimental structure-based methods. These results indicate that using pLMs as embeddings provides robustness and performance elasticity, achieving high prediction accuracy and significantly enhancing the scalability of PNBP without compromising accuracy.

Secondly, these models demonstrate excellent robustness and generalization capabilities, as evidenced by the EquiPNAS [27] experiments. GPSite [99] can predict the binding residues for ten different molecules, while GPSFun [100] is capable of annotating protein sequences with Gene Ontology (GO) terms in addition to predicting molecule interactions. As another good example, DeepProSite [50] excels in pinpointing protein–protein/peptide binding sites, and its application has been broadened to encompass the prediction of binding residues for a diverse range of ligands (e.g., Mg^2+^, Ca^2+^, and Mn^2+^) to verify its generalization ability. Results demonstrate that DeepProSite [50] outperforms its competitors across the majority of evaluation metrics, further confirming the robust and generalized capabilities of pLMs.

Thirdly, pLM embedding methods significantly accelerate PNBP by eliminating the time required to compute evolutionary information. ESM-NBR predicts a 500 bp protein sequence in just 5.52 s, approximately 16 times faster than the second-ranking DRNAPred [104], with other methods not even in the same order of magnitude. This demonstrates the clear computational speed advantage of pLMs.

Finally, the combined application of different pLMs offers complementary advantages for predicting nucleic acid binding sites, further enhancing prediction accuracy. ULDNA [51] demonstrates that not only can learned information be complementary across pre-trained models, but failures in one method’s predictions can be corrected by the other two methods. Despite the potential overlap in true positive predictions, ULDNA’s overall accuracy surpasses that of single pre-trained model methods.

### 3.5. Ablation Studies

The relative importance of features in a model can be assessed through ablation experiments, where different pLMs or different combinations with other features are formed. This allows for the investigation of the impact of various feature combinations on model performance. The ablation study allows us to further understand the contribution made by pLMs in PNBP.

Extensive experiments have demonstrated that evolutionary information has difficulty in replacing the role of pLMs in predictions. This suggests that pLMs may already encapsulate protein evolutionary information and possess richer data, thereby enhancing prediction quality [50]. Table 4 shows the classification of features taken by some of the methods. EquiPNAS [27] shows that even completely discarding evolutionary features results in only a negligible decrease in prediction accuracy for protein–nucleic acid binding sites. This underscores the importance of using pLMs as predictive embeddings, while features derived from pLMs significantly contribute to model performance.

Moreover, combining pLMs with appropriate features is not redundant but positively impacts model performance. Features such as structural features (DSSP), GO annotations, etc., are also important to improve model performance. Therefore, identifying and utilizing suitable combined features is essential.

We already know that different pLMs can complement each other in terms of information. Further ablation studies have investigated three significant pLMs: ESM-MSA, ProtTrans, and ESM2. ULDNA [51] combined these different pLMs as feature embeddings, and results showed that the inclusion of ESM2 brought the most significant performance improvement. Thus, among the three pLMs, ESM2 contributed the most.

In conclusion, feature ablation studies have demonstrated the powerful impact of pLM embeddings on PNBP. Compared to traditional feature extraction methods, pLMs can learn more effective discriminative features, reducing the reliance on conventional sequence- and structure-based features. Additionally, the embeddings from pre-trained pLMs decrease the model’s dependency on evolutionary information, enabling feature extraction for orphan proteins or rapidly evolving proteins with sparse evolutionary data. This also avoids the time-consuming task of generating MSA and PSSM features. Furthermore, even when using only pLMs, the performance of EquiPNAS [27] is comparable to or better than the current best-in-class for PNBP. This indicates that pLMs can effectively learn usable evolutionary information embedded within the protein sequences themselves. All these points illustrate that methods utilizing pLMs offer robustness and can be used to develop a versatile and scalable model, standing out against other advanced approaches.

## 4. Discussion

With the emergence of multi-million protein sequence databases, pLMs are becoming increasingly larger (e.g., ESM2 has 1.5 billion parameters). However, training such large-scale pLMs is often impractical for academic research teams. Therefore, it is advisable for academic researchers to leverage existing pre-trained language model embeddings with good generalization abilities for downstream tasks. In the case of PNBP, the pLMs are not readily usable due to the presence of the nucleic acid ligands and need to be customized to at least take the interplay between protein and the nucleic acids into account.

Moreover, it is important to note that increasing the size of pLMs does not always result in better model performance. Nijkamp et al. [105] found that larger models do not necessarily yield better zero-shot fitness performance. Similarly, ESM2 [75] points out that the improvement of small-scale pLMs tends to saturate when dealing with proteins that have a high evolutionary depth. However, for proteins with low evolutionary depth, increasing the model size significantly improves performance. This suggests that integrating appropriate biological or physical prior knowledge (e.g., PSSM, MSA, DSSP, GO) with pLMs can not only reduce the size of the pLM but also enhance the performance of downstream tasks. The involvement of multi-source data in task design implies that multi-task or multi-modal learning is worth exploring. On the other hand, the language models built upon nucleic acids, such as the DNA language model [76], RNA language model [106], and the combined biological language models [107], could be integrated into the tasks of PNBP for residue-level properties with high accuracy and efficiency. Since nucleic acids can bind to different sites on the surface of a protein, the structural features of proteins, especially local structural information, would not lose their importance if not play an increasingly important role in the near future. As RNA molecules are highly flexible, RNA language models could be very useful in guiding the related structure prediction tasks. How the local structural information can be extracted to well complement existing pre-trained pLMs is challenging. To note a promising direction, Zheng et al. proposed a multi-view graph embedding fusion of two networks that capture the global and local embedding representations, respectively [108].

In the case of PNBP, many methods have opted to create new datasets to meet the demand. This has introduced discrepancies in assessing the efficacy of varied approaches, eliciting apprehensions over potential data prejudices and ethical implications. Hence, the establishment of comprehensive, credible, and impartial benchmarks becomes imperative for assessing diverse models and promoting the advancement of dependable methodologies. Due to the existence of many approaches, a systematic assessment of these approaches on plenty of datasets could benchmark the performance and the potential problems (such as dataset bias) to guide the efforts to improve them in the coming years.

As the progress of PNBP with pLMs continues, it is of great possibility that unknown facets of protein–nucleic acid binding could be revealed and novel nucleic acid-binding proteins of desired properties could be designed. One challenge to this endeavor is that the complex embedding representation in pLMs can hardly be interpretable by any human. Understanding how protein sequences are processed and represented is crucial to identifying how models predict nucleic acid binding sites, which is helpful in protein design. On the one hand, pLMs can be linked to interpretable molecular representations such as physicochemical properties. On the other hand, model interpretability can be improved by machine learning approaches that are self-explainable [109]. In addition, biological knowledge and physical principles could be integrated within the frameworks of machine learning models to develop easily comprehensible models [110]. For most of the deep learning models in PNBP, their applicability domains are not unambiguously discussed and there are risks that such models could be applied to certain applications in which the underlying assumptions are not satisfied. Proteins are usually flexible and undergo conformational changes upon ligand binding. One is thus cautioned when applying pLMs to PNBP applications where changes in conformation and allosteric effects play a role, although successful applications of pLMs are noted in other related tasks such as drug discovery and protein engineering [111].

In addition, the role of IDP or IDR in protein–nucleic acid interactions complicates binding site prediction. Essentially, the structure of IDPs or IDRs is highly flexible and capable of dynamically interacting with nucleic acid binding. This structural plasticity allows them to participate in a wide range of binding events and often modulate binding affinity and specificity. Many intrinsic disorder predictors for IDRs that interact with proteins and nucleic acids such as DisoRDPbind [112] and DeepDISOBind [113] have been proposed and can be found, for example, in a recent survey [114]. Some of these predictors including the ones for disordered nucleic acids–binding proteins have also been accessed recently based on a novel benchmark dataset with reduced similarity to existing datasets [115]. Very recently, HybridDBRpred was developed to improve sequence-based prediction of DNA-binding residues for both the structure-annotated proteins and the disorder-annotated proteins and thus reduced prediction biases from different annotation types [116]. Both amino acid level and structural level information have been used in existing intrinsic disorder predictors; however, the utilization of pLMs in such tasks has not been reported yet. One possible reason could be that the flexibility of IDRs makes it difficult for pLMs to accurately predict binding sites, as conventional models may not fully capture the transient and dynamic nature of these interactions. Given this complexity, it is critical to develop and refine computational methods that better account for protein flexibility and the unique properties of IDPs or IDRs in nucleic acid binding. Future development of pLMs should focus on incorporating structural dynamics and disorder into their predictions to improve the accuracy and reliability of PNBP [117].

## 5. Conclusions

This paper systematically reviews the recent advancements in protein–nucleic acid binding site prediction. It covers the background, prediction challenges, the development of pLMs, and their application in this field. We highlight several successful cases that demonstrate the superior prediction quality achieved by pLM embeddings, further emphasizing the advantages of pLMs. Additionally, we discuss current limitations, potential directions, and future trends. Ultimately, we anticipate that language modeling will play a significant and convincing role in specific biological domains in the future.

## Figures and Tables

**Figure 1 genes-15-01090-f001:**
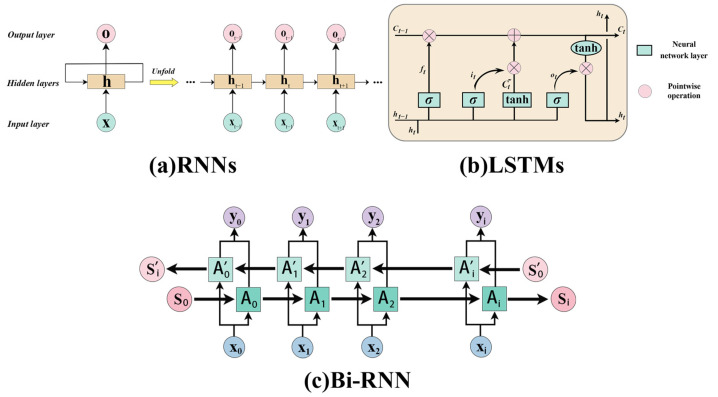
The structure of RNN, Bi-RNN, LSTM.

**Figure 2 genes-15-01090-f002:**
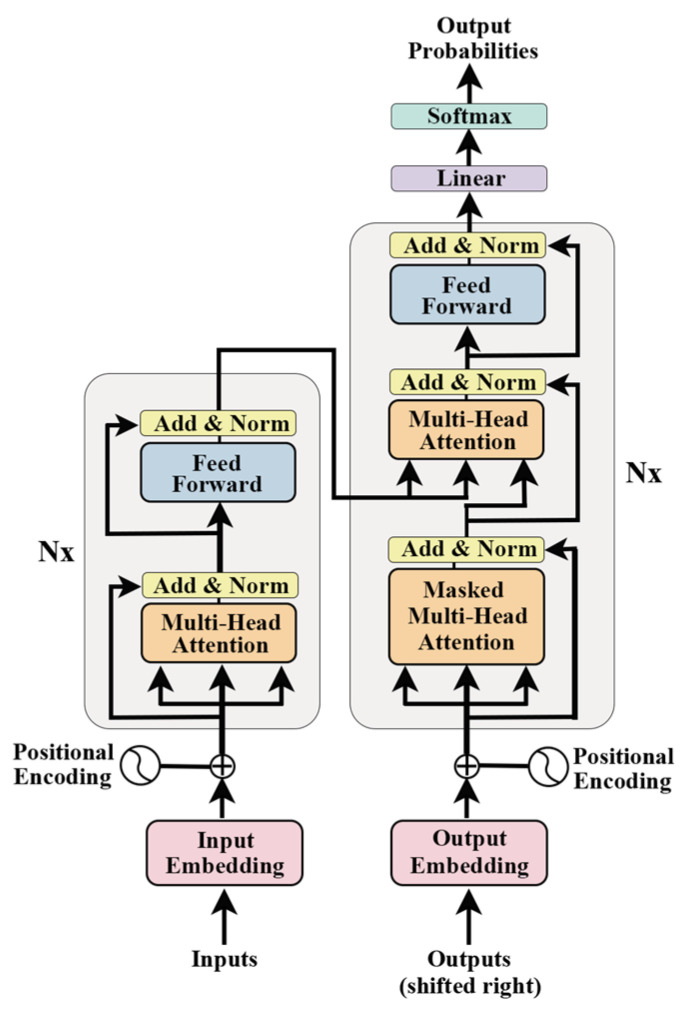
The Transformer model architecture.

**Figure 3 genes-15-01090-f003:**
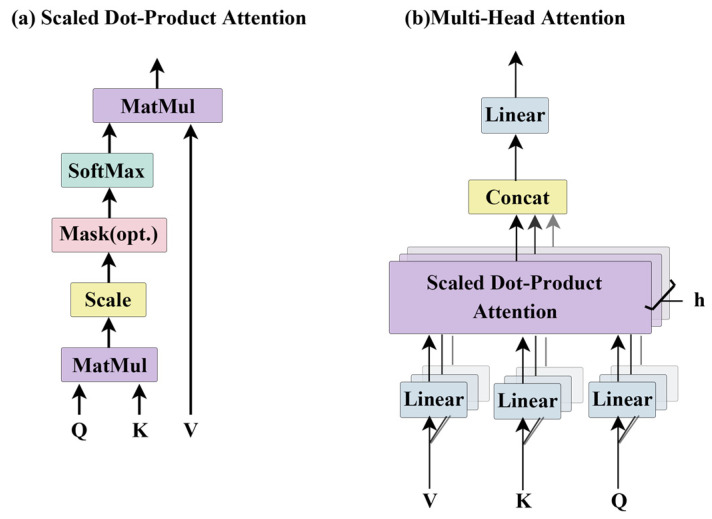
Scaled Dot-Product Attention (**a**). Multi-Head Attention consists of several attention layers running in parallel (**b**).

**Figure 5 genes-15-01090-f005:**
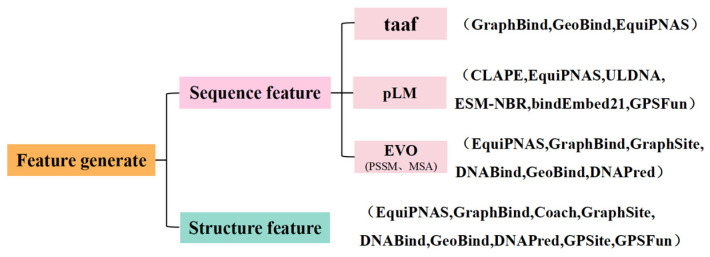
Classification of protein characteristics. At the sequence level, features are categorized into traditional amino acid features (taaf), enriched features extracted from pLM, and evolutionary features containing PSSM and MSA. At the structural level, they lack a concise classification and are elaborated in Section 3.3.3 in detail. Methods that utilize such features are listed in the parentheses. Some methods such as DNABind and GeoBind utilize both sequence features and structural ones.

**Table 3 genes-15-01090-t003:** Introduction to pLM extraction of features.

Method	Feature Generation	Feature Representation	Key Learning Architecture
CLAPE [53]	ProtBert	Tensor	Concatenated ACNNs
ESM-NBR [52]	ESM2	Tensor	LSTM
DeepProSite [50]	ProtBert, DSSP	Graphs	GNN
EquiPNAS [27]	ESM2, DSSP, PSSM, MSA, taaf, SS, RSA, et al.	Graphs	GNN
ULDNA [51]	ESM, ESM-MSA, ProtBert	Tensor	LSTM
GPSite [99]	ProtBert, ESMFold	Graphs	GNN
GPSFun [100]	ProtBert, ESMFold	Graphs	GNN

**Table 4 genes-15-01090-t004:** Feature combinations of different methods for ablation studies. Bolded text is the combination of features used in the full version of the methods.

ULDNA [51]	DeepProsite [50]	EquiPNAS [27]
ProtTrans + ESM-MSA ESM2 + ESM-MSA ESM2 + ProtTrans **ESM2 + ProtTrans + ESM-MSA (ULDNA)**	EVO DSSP ProtT5 ProtT5 + EVO + DSSP EVO + DSSP ProtT5 + EVO **ProtT5 + DSSP (DeepProSite)**	No ESM2 No (PSSM + MSA) No MSA No PSSM **ESM2, DSSP, PSSM, MSA, taaf, SS, RSA, et al. (EquiPNAS)**

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
