# Peer review of "Advances in the Application of Protein Language Modeling for Nucleic Acid Protein Binding Site Prediction"

_genes, 2024, doi:10.3390/genes15081090_

Round 1

Reviewer 1 Report

Comments and Suggestions for Authors

The paper by Wang and Li outlines the evolution of protein language models and reviews recent methods for predicting protein and nucleic acid binding sites. It covers benchmark sets, feature generation techniques, performance comparisons, and feature ablation studies, highlighting the significance of protein language models in this context. The authors conclude by discussing challenges in binding site prediction and proposing future research directions. I get the impression that the authors are quite familiar with the issues related to the prediction of protein structures, but they omit the topic of prediction of nucleic acid structures altogether. In the article we have a lot of content on protein prediction, and then on prediction of protein-nucleic acid binding sites. For completeness, the article should be supplemented with information on prediction of nucleic acid structures and language models developed for these molecules. Some language issues also appea. Other comments follow:

Major:

(1) "Notably, the realm of protein structure prediction has undergone significant advancements ..." - In the Introduction, there is a whole paragraph concerning protein structure prediction. It is fine, but there is nothing about the prediction of nucleic acids. And since the paper addresses the prediction of protein-nucleic acid binding sites, the authors should also touch nucleic acids to complete the picture of prediction. There are some good, recent papers summarizing the problems of nucleic acid structure prediction. For example, Wang et al., 2023 (doi:10.3390/molecules28145532); Li et al, 2024 (doi:10.1038/s41467-024-45191-5); Schneider et al., 2023 (doi: 10.1093/nar/gkad726); Ou et al., 2022 (doi:10.1021/acs.jcim.2c00939).

(2) When writing about the prediction of protein structures, it is worth mentioning CASP, which highly contributed to improving the accuracy of 3D structure modeling. Now, CASP also challenges the prediction of nucleic acid structures, as well as protein-NA complexes, thus, it is very much within the theme of the paper.

(3) According to the language models, please, include some information about DNABERT and RNABERT, which have been designed for nucleic acids. See Ji et al., 2021 (doi:10.1093/bioinformatics/btab083) and https://evryrna.ibisc.univ-evry.fr/evryrna/RNA-TorsionBERT

(4) Figure 2. - I suggest making it bigger (now the font is too small), and rather in a horizontal layout.

Minor:

(a) "NCBRPred[8] , DNAPred[9] , DNAgenie[10] , RNABindRPlus[11] , ProNA2020[12],ConSurf [13], TargetDNA[14] , SCRIBER[15] ,and TargetS[16] ," - spaces should be inserted after the comma, not before; please correct here and also in the other places;

Here it should be like this: "NCBRPred[8], DNAPred[9], DNAgenie[10], RNABindRPlus[11], ProNA2020[12], ConSurf[13], TargetDNA[14], SCRIBER[15], and TargetS[16],"

(b) "Moreover, the process of experimentally determining protein structures is both time-intensive and." - unfinished sentence

(c) "The output of the LSTM has two outputs" -> "The LSTM has two outputs" 

(d) The authors inconsistently apply spelling rules, for example, sometimes they put spaces before references, and sometimes they do not - this should be standardized.

(e) "Vaswani et al[48]. proposed" -> "Vaswani et al.[48] proposed"

(f) "see figure 5." -> "see Figure 5."

Comments on the Quality of English Language

Some comments on English are given above.

Author Response

I would like to thank both reviewers for their insightful and constructive suggestions, as these comments helped us to improve the quality of our work. Detailed responses to reviewers are addressed below point by point. The reviewers' comments are highlighted in italics. The modifications response to the reviewers 1 and 2 followed the comments are highlighted in blue and red, respectively, in the manuscript.

Reviewer 1

Comments and Suggestions for Authors

The paper by Wang and Li outlines the evolution of protein language models and reviews recent methods for predicting protein and nucleic acid binding sites. It covers benchmark sets, feature generation techniques, performance comparisons, and feature ablation studies, highlighting the significance of protein language models in this context. The authors conclude by discussing challenges in binding site prediction and proposing future research directions. I get the impression that the authors are quite familiar with the issues related to the prediction of protein structures, but they omit the topic of prediction of nucleic acid structures altogether. In the article we have a lot of content on protein prediction, and then on prediction of protein-nucleic acid binding sites. For completeness, the article should be supplemented with information on prediction of nucleic acid structures and language models developed for these molecules. Some language issues also appea. Other comments follow:

Reply: Thank the reviewer for the insightful comments. It summarizes the main work of the manuscript and the main directions of revision.

Major COMMENTS

  • "Notably, the realm of protein structure prediction has undergone significant advancements ..." - In the Introduction, there is a whole paragraph concerning protein structure prediction. It is fine, but there is nothing about the prediction of nucleic acids. And since the paper addresses the prediction of protein-nucleic acid binding sites, the authors should also touch nucleic acids to complete the picture of prediction. There are some good, recent papers summarizing the problems of nucleic acid structure prediction. For example, Wang et al., 2023 (doi:10.3390/molecules28145532); Li et al, 2024 (doi:10.1038/s41467-024-45191-5); Schneider et al., 2023 (doi: 10.1093/nar/gkad726); Ou et al., 2022 (doi:10.1021/acs.jcim.2c00939).
  • When writing about the prediction of protein structures, it is worth mentioning CASP, which highly contributed to improving the accuracy of 3D structure modeling. Now, CASP also challenges the prediction of nucleic acid structures, as well as protein-NA complexes, thus, it is very much within the theme of the paper.

Reply: Thank the reviewer for the constructive advice. Indeed, elaboration of the structure of nucleic acids is necessary for the prediction of protein-nucleic acid binding sites. Therefore, in the manuscript, progress in nucleic acid structure prediction is introduced after the elaboration of progress in protein structure prediction, thus naturally unfolding the account of the advantages of language modeling. In addition, the latest two years of CASP results are inserted, favorably demonstrating the feasibility of using deep learning to predict accounting binding sites (highlighted in blue, page 2 lines 78-97).

“Notably, the realm of protein structure prediction has undergone significant ad-vancements, largely fueled by the groundbreaking application of deep learning tech-niques. For example, in the structure prediction competition CASP 14[29], Al-phaFold2[30] and RoseTTAFold[31] made a major breakthrough in protein tertiary structure prediction, providing raw structural data for large-scale PNBP as a reliable alternative to experimental methods. In addition to understanding protein structure, nucleic acid structure is equally critical for elucidating the mechanisms of pro-tein-nucleic acid interactions. Accurate nucleic acid structures can reveal important binding sites and conformational changes that occur upon binding. Significant progress has been made in structure prediction by deploying large-scale pre-trained biological language models through the attention-based Transformer network. Traditional com-putational methods for nucleic acid structure prediction play a crucial role in this field. Thermodynamic models predict the secondary structure of nucleic acids based on se-quence information, calculate the minimum free energy of possible structures, and de-termine the most stable conformation. Physics-based modeling methods, on the other hand, use fragment assembly and energy minimization to predict nucleic acid structures with high accuracy. The conformational space of nucleic acid molecules can be explored through Monte Carlo simulations, thus enabling the modeling of large and complex nucleic acid structures[32–35]. In CASP 15[36], which focuses more on protein complex and RNA structure prediction, Alchemy RNA learns richer sequence information through pre-trained RNA language models(RNA-FM[37]), ranking first among all AI's methods. Besides, protein language modeling (pLM) has also achieved great results. ESMFold[33], for example, differs from previous methods by generating po-sition-specific scoring matrices (PSSMs) from multiple sequence alignments (MSAs) using only protein sequences as inputs. This improves the speed of prediction while maintaining high accuracy at the atomic level. In summary, ESMFold[33] surpasses other methods in handling proteins with limited homologous sequences. Besides pro-tein structure prediction, there is evidence that pLMs also perform well in various other predictive modeling tasks, including protein function annotation[34,35], protein de-sign[36,37], and ligand binding prediction[38,39]. This undoubtedly indicates that pLMs have significant potential in the downstream study of protein function and structure. Consequently, numerous researchers are now devoting their efforts to leveraging the capabilities of pLMs for the large-scale and accurate prediction of protein-nucleic acid binding sites.”

  • According to the language models, please, include some information about DNABERT and RNABERT, which have been designed for nucleic acids. See Ji et al., 2021 (doi:10.1093/bioinformatics/btab083) and https://evryrna.ibisc.univ-evry.fr/evryrna/RNA-TorsionBERTSPECIFIC COMMENTS

Reply: Thank you for your valuable input. We followed the suggestion and  have now described the protein language model and the nucleic acid language model separately in Chapter II. Specifically, on page 9, lines 295-324, the characteristics of the nucleic acid language model and its disadvantages and room for development compared to the protein language model are elaborated.

“2.4. Nucleic Acid language models

In addition to pLM, there have been good advances in language models designed specifically for nucleic acids, including DNABERT[76] and RNA-TorsionBERT[77]. These models are based on the BERT[60] (Bidirectional Encoder Representation from Transformers) architecture and are tailored to capture the unique features and sequence patterns of DNA and RNA.

DNABERT[76] and RNA-TorsionBERT[77] adapt BERT models for DNA and RNA sequences. It is trained on large-scale genomic data to learn the underlying patterns of nucleotide sequences. These models have been successfully applied to a variety of tasks and have helped to deepen the understanding of the PNBP mechanism. It provides a powerful approach to understand the complex dynamics of protein-nucleic acid inter-actions. For instance, DNABERT has been used in tasks like identifying transcription factor binding sites and predicting methylation patterns, while RNA-TorsionBERT has been applied to understand RNA conformational dynamics and to predict RNA-protein interactions.

However, in comparison to pLMs, nucleic acid language models have limited training data, which restricts the generalization ability of the models. In addition, the progress of pLMs is due to the rich evolutionary information contained in the protein sequences themselves, whereas nucleic acids themselves may not contain similarly rich information. Especially in non-coding regions and species-specific regulatory elements, it is difficult for nucleic acid language models to get better access to evolutionary in-formation[35]. Nucleic acid language modeling is at an early stage of development and needs further validation, but it still has a very promising future.

Looking forward, the future of nucleic acid language models is undoubtedly promising. Continued advancements in genomic sequencing technologies and the ac-cumulation of more comprehensive datasets could potentially address the current lim-itations. Additionally, integrating nucleic acid models with other types of biological data, such as epigenetic marks, chromatin accessibility, and transcriptional activity, could enhance their ability to make accurate predictions. As these models evolve, they are expected to play an increasingly crucial role in decoding the intricacies of genetic regulation, gene expression, and the broader mechanisms underlying protein-nucleic acid interactions.”

  • Figure 2. - I suggest making it bigger (now the font is too small), and rather in a horizontal layout.

Reply: Thanks for the suggestion, changes have been made in the manuscript.

 Minor COMMENTS

(a) "NCBRPred[8] , DNAPred[9] , DNAgenie[10] , RNABindRPlus[11] , ProNA2020[12],ConSurf [13], TargetDNA[14] , SCRIBER[15] ,and TargetS[16] ," - spaces should be inserted after the comma, not before; please correct here and also in the other places;

Here it should be like this: "NCBRPred[8], DNAPred[9], DNAgenie[10], RNABindRPlus[11], ProNA2020[12], ConSurf[13], TargetDNA[14], SCRIBER[15], and TargetS[16],"

(b) "Moreover, the process of experimentally determining protein structures is both time-intensive and." - unfinished sentence

(c) "The output of the LSTM has two outputs" -> "The LSTM has two outputs" 

(d) The authors inconsistently apply spelling rules, for example, sometimes they put spaces before references, and sometimes they do not - this should be standardized.

(e) "Vaswani et al[48]. proposed" -> "Vaswani et al.[48] proposed"

(f) "see figure 5." -> "see Figure 5."

Reply: Thank you for your careful pointing out of the problems, the statements in the manuscript have been corrected and the formatting issues have been standardized.

Reviewer 2 Report

Comments and Suggestions for Authors

It is an interesting review on “Protein Language Modeling for Nucleic Acid Protein Binding Site Prediction” by Wany and Li. I would recommend this review after addressing the concerns and suggestions follow below.

1. The author hasn’t mentioned intrinsically disordered protein (IDP) or region (IDR) anywhere in the review, specifically in the introduction section. Many studies have demonstrated the roles of intrinsic disorder in protein-nucleic Acid Interactions (see review Mol Biosyst. 2012 Jan; 8(1): 97–104.). It would be useful for other readers if author include disorder protein.

2.     In Figure5, DNABind is highlighted in both structural and sequence feature, however in the section 3, there is no clearer explanation, and DNABind is mentioned only in subsection 3.3.3 (page 12, line 423). Is DNABind feature common to both structure and sequence?

3.     Another, “Features based on amino acid” mentioned in subsection 3.3.1 (page 10, line 350) is the same as “Features based on sequence” shown in Figure 5? If yes, then author need to maintain the consistency to avoid confusion.

4.     I would also suggest author to include “BioLiP” in Figure 5 in addition to GraphBind datasets, because most of the discussion are based on BioLiP.   

5.     Minor: In Figure 5, under “EVO”, DNABindGeoBind should be separated by comma (,).

Author Response

Reviewer 2

Comments and Suggestions for Authors

It is an interesting review on “Protein Language Modeling for Nucleic Acid Protein Binding Site Prediction” by Wany and Li. I would recommend this review after addressing the concerns and suggestions follow below.

Reply: Thank you for your positive comments.

Major COMMENTS

  1. The author hasn’t mentioned intrinsically disordered protein (IDP) or region (IDR) anywhere in the review, specifically in the introduction section. Many studies have demonstrated the roles of intrinsic disorder in protein-nucleic Acid Interactions (see review Mol Biosyst. 2012 Jan; 8(1): 97–104.). It would be useful for other readers if author include disorder protein.

Reply: Thank you for your advice. After learning more about IDP and IDR, in the manuscript we made the necessary clarifications.IDP/IDR illustrates the importance of nucleic acid binding site prediction methods due to the absence of stable three-dimensional structures. (highlighted in red, page 1 lines 40-44).

“In addition, studies in recent years have emphasized about the key role of intrinsically disordered protein (IDP) or region (IDR) in protein-nucleic acid interactions. This in-cludes RNA maturation, ribosome assembly, etc[7,8]. Unlike structural proteins, IDPs and IDRs lack a fixed three-dimensional structure under physiological conditions. This is also difficult to study by means of experimental assays.”

The introduction of IDP/IDR in the manuscript, in the section on the current remaining problems of computational methods, is used to emphasize the inevitability of the development of new efficient forecasting methods.(page 2 lines 70-71)

“However, these methods perform poorly when predicting orphan proteins that lack similar entries in the database. The extraction of evolutionary features from proteins necessitates a substantial investment of time. IDP and IDR bring unique challenges for PNBP due to their lack of stable structure and high sequence variability[7]. Lastly, it is crucial to note that current methodologies heavily rely on manually curated features to encapsulate structural information and construct predictive models. This approach requires extensive domain knowledge and may fail to capture essential biological features for specific tasks”

In the discussion section of the manuscript, we added the impact of IDR on PNBP and the way forward.(page 16 line 632-642)

“In addition, the role of IDP or IDR in protein-nucleic acid interactions complicates binding site prediction. Essentially the structure of IDPs or IDRs is highly flexible and capable of dynamically interacting with nucleic acid binding. This structural plasticity allows them to participate in a wide range of binding events and often modulate bind-ing affinity and specificity. Many intrinsic disorder predictors for IDRs that interact with proteins and nucleic acids such as DisoRDPbind[112] and DeepDISOBind[113] have been proposed and can be found, for example, in a recent survey[114]. Some of these predictors including the ones for disordered nucleic acids-binding proteins had also been accessed recently based on a novel benchmark dataset with reduced similar-ity to existing datasets[115]. Very recently, HybridDBRpred was developed to improve sequence-based prediction of DNA-binding residues for both the structure-annotated proteins and the disorder-annotated proteins and thus reduced prediction biases from different annotation types[116]. Both amino acid level and structural level information has been used in existing intrinsic disorder predictors, however, the utilization of pLMs in such tasks has not been reported yet. One possible reason could be that the-flexibility of IDRs make it difficult for pLMs to accurately predict binding sites, as conventional models may not fully capture the transient and dynamic nature of these interactions. Given this complexity, it is critical to develop and refine computational methods that better account for protein flexibility and the unique properties of IDPs or IDRs in nucleic acid binding. Future development of pLMs should focus on incorpo-rating structural dynamics and disorder into their predictions to improve the accuracy and reliability of PNBP[117].”

  1. In Figure5, DNABind is highlighted in both structural and sequence feature, however in the section 3, there is no clearer explanation, and DNABind is mentioned only in subsection 3.3.3 (page 12, line 423). Is DNABind feature common to both structure and sequence?

Reply: Thank you for your question. First of all, DNABind as a method of predicting nucleic acid binding sites uses both structural and sequence level features. Secondly, the features mentioned in this section of 3.3 are generic, so the manuscript only uses individual methods as examples. This is also the reason why DNABind only appears in one section. Finally Figure 5 is also just a summary of the feature dimensions taken by the different methods. We have modified Figure 5 and its caption to eliminate potential ambiguities. (page 9 line 306-313)

“Figure 5 categorizes the features that are currently in common use, and the methods mentioned in the text involve a wide range of features at the sequence and structural level.

Figure 5. Classification of protein characteristics. At the sequence level, features are categorized into tra-ditional amino acid features(taaf), enriched features extracted from pLM, and evolutionary fea-tures containing PSSM, MSA. At the structural level they lack a concise classification and are elaborated in Sec. 3.3.3 in detail. Methods that utilize such features are listed in in the parenthesis. Some methods such as DNABind and GeoBind utilize both sequence feature and structural ones.”

  1. Another, “Features based on amino acid” mentioned in subsection 3.3.1 (page 10, line 350) is the same as “Features based on sequence” shown in Figure 5? If yes, then author need to maintain the consistency to avoid confusion.

Reply: We really thank the reviewer for these helpful suggestions. "Features based on amino acids" is different from "Features based on sequence". To avoid confusion, we renamed “Features based on amino acid” into traditional amino acid features(taaf) in Figure 5.

  1. I would also suggest author to include “BioLiP” in Figure 5 in addition to GraphBind datasets, because most of the discussion are based on BioLiP.   

Reply: We are sorry for this misunderstanding due to the lack of clarity for Figure 5 in the previously version of the caption. Now, it is clear that Figure 5 categorizes the features instead of methods. GraphBind utilize both structural and sequence features and thus is listed in both categories.  As BioLip is a dataset and thus it is not included in Figure 5.

Minor COMMENTS

5. In Figure 5, under “EVO”, DNABindGeoBind should be separated by comma (,).

Reply: We have corrected the Figure 5.

Round 2

Reviewer 1 Report

Comments and Suggestions for Authors

The authors responded to all my comments and questions. I have no further remarks.